# Modeling a Fluid-Coupled Single Piezoelectric Micromachined Ultrasonic Transducer Using the Finite Difference Method

**DOI:** 10.3390/mi14112089

**Published:** 2023-11-12

**Authors:** Valentin Goepfert, Audren Boulmé, Franck Levassort, Tony Merrien, Rémi Rouffaud, Dominique Certon

**Affiliations:** 1GREMAN UMR7347, CNRS, INSA CVL, University of Tours, 37100 Tours, France; valentin.goepfert@etu.univ-tours.fr (V.G.); franck.levassort@univ-tours.fr (F.L.); remi.rouffaud@univ-tours.fr (R.R.); 2MODULEUS SAS, 37100 Tours, France; audren.boulme@moduleus.com (A.B.); tony.merrien@outlook.fr (T.M.)

**Keywords:** PMUT, ultrasound, MEMS, finite difference method, characterization, lumped-element, vibrometry

## Abstract

A complete model was developed to simulate the behavior of a circular clamped axisymmetric fluid-coupled Piezoelectric Micromachined Ultrasonic Transducer (PMUT). Combining Finite Difference and Boundary Element Matrix (FD-BEM), this model is based on the discretization of the partial differential equation used to translate the mechanical behavior of a PMUT. In the model, both the axial and the transverse displacements are preserved in the equation of motion and used to properly define the neutral line position. To introduce fluid coupling, a Green’s function dedicated to axisymmetric circular radiating sources is employed. The resolution of the behavioral equations is used to establish the equivalent electroacoustic circuit of a PMUT that preserves the average particular velocity, the mechanical power, and the acoustic power. Particular consideration is given to verifying the validity of certain assumptions that are usually made across various steps of previously reported analytical models. In this framework, the advantages of the membrane discretization performed in the FD-BEM model are highlighted through accurate simulations of the first vibration mode and especially the cutoff frequency that many other models do not predict. This high cutoff frequency corresponds to cases where the spatial average velocity of the plate is null and is of great importance for PMUT design because it defines the upper limit above which the device is considered to be mechanically blocked. These modeling results are compared with electrical and dynamic membrane displacement measurements of AlN-based (500 nm thick) PMUTs in air and fluid. The first resonance frequency confrontation showed a maximum relative error of 1.13% between the FD model and Finite Element Method (FEM). Moreover, the model perfectly predicts displacement amplitudes when PMUT vibrates in a fluid, with less than 5% relative error. Displacement amplitudes of 16 nm and 20 nm were measured for PMUT with 340 µm and 275 µm diameters, respectively. This complete PMUT model using the FD-BEM approach is shown to be very efficient in terms of computation time and accuracy.

## 1. Introduction

Piezoelectric Micromachined Ultrasonic Transducers (PMUTs) are MicroElectroMechanical Systems (MEMSs) whose working principle is comparable to that of a microphone. A single PMUT consists of a plate or membrane whose operating frequency depends on its geometrical parameters (i.e., size, shape, and thickness) which can be tuned according to transducer specifications [1]. Actuation or electromechanical coupling is provided by a piezoelectric material deposited as a thin film (i.e., typically micrometric thicknesses) on the top of the plate using processes derived from microelectronic manufacturing techniques. These devices work like Capacitive Micromachined Ultrasonic Transducers (CMUTs) [2,3] except for the actuation which is based on electrostatic forces. The first relevant proof-of-concept publications related to these technologies date back to the early 2000s, with two articles showing functional PMUT transducers, one based on Zinc Oxide (ZnO) [4,5] and the other on Lead Zirconate titanate (PZT) [1,6]. Compared with ultrasonic transducers based on bulk ceramic, PMUT technology offers several benefits that demonstrate why, for more than twenty years, there has been continually growing research and development dedicated to these technologies. The main advantages of such technology are the ability to address high-volume markets, rapid manufacturing times, easier integration with electronics, reduced production costs, and miniaturization of ultrasound probes. While we do not summarize all the work carried out over the last 20 years, as proposed in [7], three main categories of results are identified: materials and manufacturing processes, applications, and modeling.

In the field of materials and processes, recently published articles [7,8,9,10,11] provide an exhaustive review of all the technological developments to date. The most widely used piezoelectric materials are the following: ZnO [4,12,13]; AlN, for which scandium doping [14,15] has enabled the electroacoustic performances to be three times higher; and PZT material, the first works on which were published by [6,16,17,18] and, more recently, by Savoia et al. [19]. Thin-film PZT is usually deposited either by a sol–gel process [20] or a sputtering technique [20]. There are two main types of manufacturing processes depending on one’s desire or ability to make the device cavities. On the one hand, the backside etching techniques require, once the structural layer has been deposited, the release of the membranes by Deep Reactive Ion Etching (DRIE) on the backside of the substrate. On the other hand, using surface micromachining processes, cavities are created by wet chemical etching or wet vapor etching [21,22]. This process approach requires the cavities to be sealed to prevent liquid or gas infiltration. Cavities can be made by dry etching and then sealed using a wafer bonding technique where a Silicon On Insulator (SOI) is used to make the structural layer of the membranes. In this case, the piezoelectric material is deposited after the bonding step [23].

Among the broad range of PMUT application domains, two main categories stand out: ultrasonic air-coupled and ultrasonic fluid-coupled applications. The first domain involves applications such as telemetry, with the manufacture of dedicated electronic components [24,25] and gesture or movement recognition [26,27]. The second domain is mainly dedicated to medical applications, including ultrasound imaging and high-intensity therapeutic ultrasound capabilities [28,29].

On the modeling side, many studies have been conducted to propose models that can predict the performance of a PMUT, whether coupled to air or fluid. Despite the wide diversity of works, the theoretical approaches used can be classified into two main categories: those based on the use of commercial finite element codes and those based on an analytical resolution of the equations governing PMUT arrays. The use of finite elements is mandatory to simulate the response of complex-shaped structures [30,31,32] and to help the development of analytical models [27,33,34], with the aim of validating or verifying initial hypotheses [35,36]. In addition, we should mention the recent work of Savoia et al. [19], who developed a complete electroacoustic model of a PMUT array element. Although highly reliable, the use of finite element models requires significant computing resources and calculation times that can become long if the acoustic boundary conditions do not allow for a reduction in the number of PMUT cells to be considered in the simulation. For example, to simulate an array element, each PMUT must be considered individually because in some cases, the periodicity or symmetry conditions cannot be applied [37]. The use of analytical models thus remains essential, as they generally enable many configurations to be simulated quickly. Most analytical models are based on two main steps: the first step is to solve the equations of a single PMUT cell and to define its equivalent lumped-element circuit model. The second step is to model the array including neighboring PMUTs. Usually, this step is achieved by computing the Boundary Element Matrix (BEM), which enables acoustic coupling of all the PMUTs. The terms of this matrix are the self and mutual acoustic radiation impedances of each PMUT [38,39,40], which are typically calculated analytically. This approach was initially proposed by our group for CMUT transducers [41], as did many other authors later [42,43,44,45]. As a result, the determination of mutual acoustic impedances is now well established, both for PMUT and CMUT technologies. Numerous transducer models have been introduced, depending on the shape and the source condition used for membrane deformation. The key element of the analytical models therefore remains the modeling of the single cell and the determination of its equivalent electromechanical scheme when coupled to a fluid medium.

The purpose of this paper is to develop an equivalent lumped-element circuit model of a circular single PMUT cell coupled to a fluid medium based on a finite difference (FD) calculation approach to solve the mechanical equations of the plate. Despite numerous modeling publications to date [33,34,35,46,47,48], we conclude that this work offers novel contributions based on many solutions having been proposed based on hypotheses that were not rigorously validated. In parallel to the development of our model, we compare this work with that of the literature by discussing, at each step, the validity of various simplifying assumptions that are usually made. Among the key contributions of this paper, we discuss and quantify multiple parameters, methods and hypotheses that have been commonly employed for previous PMUT modeling:The choice of the membrane Young’s modulus value for a disc-shaped PMUT when the structural material is anisotropic (e.g., silicon);The method to determine the neutral line while solving the plate equation that governs membrane deformation [49,50,51,52];The impact of the electrical neutral line as introduced by Samourra et al. [35,49] on the electromechanical PMUT response;The equivalent lumped-element circuit components’ calculation according to the model chosen, i.e., Foldy’s model or Mason’s model.

The remainder of this paper is structured as follows. Section 2 focuses on the resolution of the PMUT behavioral equations when the membrane is acoustically coupled with a fluid medium. Here, the plate equation is employed with minimal assumptions, and we propose an original method to separate the out-of-line deformation equation from the in-plane deformation equation. We use the method described in [47,50], which proposes separating the equations by introducing a reference line rather than a neutral line. To model the PMUT/fluid coupling, we use a Green’s function dedicated to the radiation of axisymmetric transducers combined with a boundary element matrix as presented in the model developed by Meynier et al. [41]. We show that this approach, unlike the classical analytical literature models, enables the high cutoff frequency of the PMUT to be predicted accurately. This high cutoff frequency is of great importance, as it defines the upper limit that corresponds to the situation where the spatial average velocity of the plate is null. For all the simulations, the equations are solved numerically using a FD discretization scheme [53].

Section 3 introduces electromechanical coupling through the piezoelectricity equations. As discussed on several occasions by Sammoura et al., [35,36,49] while the reference line enables us to simplify the mechanical equations, it is not sufficient for the electrical part to separate equations of the in-plane mechanical displacement from that of the out-of-plane displacement. Nevertheless, once the equations have been completely solved, we show that the calculations can be simplified without introducing significant errors in the accuracy of the results so that only the out-of-plane displacement components remain as degrees of freedom. As in the first part, all the equations are solved using the FD method. In addition, the equivalent electrical circuit model of a single PMUT cell coupled to a fluid medium will be set up.

Finally, Section 4 is dedicated to the experimental validation of the model developed using circular AlN-based devices of various sizes. The model is compared with the experimental results using several series of electrical and laser interferometry measurements taken when the PMUT vibrates in air and in oil. We note that while many results have already been published for air-coupled PMUTs, it is much more unique to report experimental results for fluid-coupled PMUTs.

## 2. Acoustic and Mechanical Modeling of a Single PMUT Cell Coupled with a Fluid Medium

As stated, this section is devoted to the resolution of the mechanical equations that govern the PMUT plate, in air and when the plate is coupled with a fluid medium. Mechanical equations are first developed, described, and then solved by means of an FD discretization scheme. Second, a first model validation will be presented by comparing the simulated resonance frequencies with those provided by a commercial finite element code (COMSOL Multiphysics [54,55]). Then, the fluid/PMUT coupling equations will be introduced in numerical form, after the FD discretization step. The last subsection is the second validation step of the model, where simulated results are compared with analytical models from the literature.

Before providing the mechanical plate description, the typical geometry of a PMUT cell with the corresponding coordinate system axis is established. Our PMUT structure plate is made up of three layers (Figure 1) and is axisymmetric. The layers include a structural layer, usually silicon, a piezoelectric layer, and a partially covering top electrode. To be consistent with the experimental devices presented in the last section, there is no metallic bottom electrode; because the silicon material is doped, it also plays the role of an electrode. The plate is considered to be mechanically clamped around its outer perimeter. In accordance with the PMUT circular geometry, the plate analysis will be carried out in cylindrical coordinates (r,θ,z). The terms a and $r_{e}$ are the plate radius and the top electrode radius, respectively. The position of the top of the *k*-indexed layer along the *z*-axis is referenced with the $h_{k}$ coordinate, where $h_{0}=0$, and ($h_{k}$ − $h_{k-1}$) corresponds to the thickness of the *k*-layer. Note that to clearly identify the piezoelectric layer from the others, its upper coordinate is labeled $h_{p}$. The dashed line with coordinate $z_{n}$ is the reference line, the so-called neutral line, whose expression and value will be given later.

For all calculations, each *k*-layer is assumed to be isotropic and mechanically characterized by its Young’s modulus $E_{k}$ and its Poisson’s ratio $\sigma_{k}$.

### 2.1. Mechanical Behavioral Equations and Resolution with Finite Difference Discretization

The mechanical behavior of the PMUT plate is based on the resolution of the classical Kirchhoff–Love thin plates equation [56,57,58] expressed under cylindrical coordinates [46,48,59,60]. This theory relies on two mains assumptions:(1)The mechanical plate vibration is limited to the displacements u(r,z) and w(r,z) along the *r*-axis (axial displacement) and z-axis (transverse displacement), respectively.(2)The through-the-thickness stresses and strains are negligible.

Total displacements u and w are shown in Figure 2. Using the physical neutral line concept and classical plate theory, the displacements take the following forms [47,58]:(1)$$u\left(r,z,t\right)=u_{0}-\left(z-z_{n}\right)\frac{𝜕w}{𝜕r},$$
(2)$$w\left(r,z,t\right)=w\left(r,t\right)=w,$$
where $u_{0}$ is the axial displacement of the reference line $z_{n}$, i.e., the neutral line, which will not be considered negligible. In many papers [35,47,61,62], authors deliberately neglect this axial displacement, which enables them to strongly simplify the analytical calculations. However, in papers from Sammoura’s group [49,63], it was shown that the axial displacement of the plate plays a role in the piezoelectric equation of the PMUT and cannot be suppressed. Therefore, at this step of the theoretical developments, we chose to keep this axial displacement in the set equations and later discuss its influence on the PMUT behavior. Due to axisymmetric deformation, only the strains $S_{rr}$ and $S_{\theta \theta }$ must be considered, and the shear strain $S_{r\theta }$ is null. Because axial displacement is not suppressed from the equations, as presented by J.N. Reddy in chapter 5 [58], to simplify equation reading, two additional matrix terms are introduced, the membrane strain matrix $\varepsilon^{0}$ and the flexural strain matrix, known as the curvature matrix $\varepsilon^{1}$:ε0=𝜕u0𝜕ru0r
ε1=−𝜕2w𝜕r2−1r𝜕w𝜕r

Using the set of Equations (1) and (2), $S_{rr}$ and $S_{\theta \theta }$ are expressed as follows:(3)SrrSθθ=ε0+z−znε1,

To establish the dynamic equilibrium equation, we classically introduce the resultant bending moment terms from a side and the resultant radial forces terms from another side as explained by J.N. Reddy [56,57,58]. Detailed description of calculations is given in Appendix A and Appendix B.

The equations of motion of the plate for circular vibrations, without an external source, are given by J.N. Reddy [58]; for the resultant moments, we have:(4)$$\frac{1}{r}\frac{{𝜕}^{2}\left(rM_{rr}\right)}{𝜕r^{2}}-\frac{1}{r}\frac{𝜕M_{\theta \theta }}{𝜕r}=\rho_{s}\frac{{𝜕}^{2}w}{𝜕t^{2}}-q\left(r\right).$$
where *M_rr_* and *M_θθ_* are the two resultant bending moments.

For the resultant stresses (*N_rr_* and *N_θθ_*), if there are no external forces acting in the volume of the plate, we have:(5)$$-\frac{1}{r}\left(\frac{𝜕\left(rN_{rr}\right)}{𝜕r}-N_{\theta \theta }\right)=-\rho_{s}\frac{{𝜕}^{2}u_{0}}{𝜕t^{2}},$$
where $\rho_{s}=\sum_{k}\rho_{k}h_{k}$ is the plate weight per unit area. Note that Equation (4) assumes that the rotary inertia terms are neglected, as explained by J.N. Reddy [58]. $q\left(r\right)$ is a source term that corresponds to an external distributed pressure that is applied to the plate.

Expressions of $M_{rr}$ and $M_{\theta \theta }$ are introduced in Equation (4) to obtain:(6)$$\frac{{𝜕}^{2}{M_{rr}}^{0}}{𝜕r^{2}}+\frac{2}{r}\frac{𝜕{M_{rr}}^{0}}{𝜕r}-\frac{1}{r}\frac{𝜕{M_{\theta \theta }}^{0}}{𝜕r}+\frac{{𝜕}^{2}{M_{rr}}^{1}}{𝜕r^{2}}+\frac{2}{r}\frac{𝜕{M_{rr}}^{1}}{𝜕r}-\frac{1}{r}\frac{𝜕{M_{\theta \theta }}^{1}}{𝜕r}=\rho_{s}\frac{{𝜕}^{2}w}{𝜕t^{2}}-q(r).$$

The first three terms on the left-hand side of Equation (6) depend on $u_{0}$, and the three others depend on w. If the axial displacement of the neutral line $u_{0}$ was neglected, Equation (6) would be a function of only the transverse displacement w. A technique to separate transverse and axial displacement components is to modify the expression of these first three terms as follows:(7)$$\frac{{𝜕}^{2}M_{rr}^{0}}{𝜕r^{2}}+\frac{2}{r}\frac{𝜕M_{rr}^{0}}{𝜕r}-\frac{1}{r}\frac{𝜕M_{\theta \theta }^{0}}{𝜕r}=\left(\frac{{𝜕}^{3}u_{0}}{𝜕r^{3}}+\frac{2}{r}\frac{{𝜕}^{2}u_{0}}{𝜕r^{2}}-\frac{1}{r}\frac{𝜕\left(\frac{u_{0}}{r}\right)}{𝜕r}\right)\left(B_{11}-z_{n}A_{11}\right).$$

If the neutral line coordinate *z_n_* is chosen such that:(8)$$B_{11}-z_{n}A_{11}=0\;\mathrm{and}\;\mathrm{so}\;z_{n}=\frac{B_{11}}{A_{11}},$$
then the terms with $u_{0}$ are removed from Equation (6).

The neutral line is the position to uncouple the axial displacement to the transverse displacement and the exact expression of $z_{n}$ is then deduced:(9)$$z_{n}=\frac{\sum_{k}\frac{h_{k}^{2}-h_{k-1}^{2}}{2}\frac{E_{k}}{1-\sigma_{k}^{2}}}{\sum_{k}\left(h_{k}-h_{k-1}\right)\frac{E_{k}}{1-\sigma_{k}^{2}}}.$$

We find the expression used in the literature [49,62], but it has never been demonstrated or even verified in the case of acoustic MEMSs. Only Sammoura proposed a demonstration [49], but the initial hypothesis of his model considered the displacement $u_{0}$ to be negligible. The same result is obtained in Appendix B for the second equation of motion.

The final equation of motion becomes:(10)$$\nabla^{2}\left(D_{eq}\nabla^{2}w\right)+\frac{{𝜕}^{2}}{𝜕r^{2}}\left(D_{eq}\left(\sigma_{eq}-1\right)\right)\frac{1}{r}\frac{𝜕w}{𝜕r}+\frac{𝜕}{𝜕r}\left(D_{eq}\left(\sigma_{eq}-1\right)\right)\frac{1}{r}\frac{{𝜕}^{2}w}{𝜕r^{2}}=\rho_{s}\frac{{𝜕}^{2}w}{𝜕t^{2}}-q(r),$$
where $\nabla^{2}=\frac{{𝜕}^{2}}{𝜕r}+\frac{1}{r}\frac{𝜕}{𝜕r}$ is the Laplacian operator in cylindrical coordinates. $D_{eq}$ and $\sigma_{eq}$ are the equivalent flexural rigidity and the equivalent Poisson’s coefficient of the plate, respectively, with their expression presented by J.N. Reddy [58].

The solution of Equation (10) can be obtained with analytical developments. In previous reports, it is common to decompose the solution into a basis of eigenmodes and to determine the coefficients of this decomposition to meet the boundary conditions. We did not choose this approach because the function basis used for air coupling does not properly account for when the PMUT plate is coupled with a fluid medium (see Section 2.3).

To solve this equation, the FD discretization scheme presented in [57] was used. Our group previously used this scheme for circular-shaped CMUTs [53]. To convert Equation (10) from its analytical to its numerical form, we first need to set up discretization nodes (see Figure 3) along the plate: from the center at *r* = 0 to the edge of the plate at *r* = a. The pitch between two nodes Δ*r* is fixed, and the number of nodes is labeled N. At each discretization node *i*, Equation (10) is combined with the discretization scheme given by Timoshenko [57] to obtain a relation between the displacement $w_{i}$ of the *i*-node with its neighboring nodes. Particular attention is given to the first node and the last node to take into account the boundary conditions. In our case, the following boundary conditions were used:(a)$r=0;\;{\left.\frac{𝜕w}{𝜕r}\right|}_{r=0}=0$, symmetrical boundary conditions,(b)$r=a;\;w\left(a\right)=0,\;{\left.\frac{𝜕w}{𝜕r}\right|}_{r=a}=0\;$, clamped boundary conditions.

The set of Equations obtained is then gathered to express the Equation (10) in matrix form:(11)$$\left[K_{m}\right]\left[w\right]=\omega^{2}\left[M\right]\left[w\right]-\left[q\right],$$
where $\left[w\right]$ is the vector made with all degrees of freedom $w_{i}$; in air, there is no external pressure, so [*q*] is null. $\left[K_{m}\right]$ is the stiffness matrix that results from the discretization of the differential operator of Equation (10), and $\left[M\right]$ is the mass matrix. $\left[M\right]$ is a pure diagonal matrix that contains the value of $\rho_{s}$ at each discretization node. The stiffness, mass, and neutral line associated with the nodes before and after the boundary are thus perfectly defined.

### 2.2. Comparison with the Finite Element Model (FEM)

The first validation step was to compare the simulation results of the FD model with those provided by a commercial finite element model. COMSOL Multiphysics was used. There are two key points to check. The validation of the neutral line coordinate $z_{n}$ computation in the case where the plate thickness is not homogeneous is the first concern. In other words, is the model able to predict the plate response when the neutral line position is discontinuous? To overcome this difficulty, some authors have proposed separating the plate into two zones and combining the solutions by applying continuity conditions at the interface between the two zones [61,64,65,66]. However, even if analytical solutions are developed, the model becomes quite “heavy” as it introduces many additional variables.

The anisotropy of the structural layer is the second concern of this validation step. All equations were developed in the case of axisymmetric vibration, assuming that the materials are isotropic. In practice, the structural material is often silicon material, which is anisotropic. As silicon is an anisotropic material [67], the Young’s modulus may vary from 130 GPa to 188 GPa according to the axis [110] or [$\bar{1}00$] in crystal lattices. In a previous study [68], it was shown that a circular isotropic silicon plate with a Young’s modulus of 143 GPa and a Poisson’s ratio of 0.278 gives the same deformation and resonance frequency as a circular anisotropic silicon plate, with less than 0.3% error. To conduct simulations, it was chosen to model silicon material as an isotropic material and to keep these elastic parameter values.

To compare the FD model with COMSOL Multiphysics, we defined a PMUT cell comparable to the literature structures based on aluminum nitride (AlN), because the devices used experimentally are also AlN-based (see Section 4). The PMUT cell is made of three layers of only doped silicon, which is the structural layer and the bottom electrode, an AlN layer, and an aluminum (Al) layer for the top electrode. The layer thicknesses and PMUT diameter were fixed to obtain a resonance frequency in air and water in the MHz range. To be consistent with the literature [22,31,35], the PMUT diameter was fixed at 50 µm. The properties of AlN used were extracted from the literature [69] and from the experimental results presented in the last section of the paper, Section 4. All material properties and dimensions are reported in Table 1, but only the mechanical properties were used for this comparison. For the FD model, 50 nodes of discretization were used along the *r*-axis, corresponding to a Δ*r* value of 1 µm. This value was fixed after comparing convergence of the resonance frequency value (2.5 MHz) for a node number varying from 10 to 100. Results showed that above 50 nodes, relative variations were lower than 0.5% and the computing time remained lower than 1 s.

For the FEM model, no asymmetrical boundary conditions were applied to the silicon layer, and the real elastic properties (anisotropic) [67] were used. The AlN and aluminum layers are not difficult to simulate because AlN has hexagonal crystal symmetry and aluminum is isotropic. The mesh of the studied structure was carefully chosen to meet the λ/4 criterion in the whole frequency range so 125 hexahedron-shaped elements in total were chosen.

Figure 4 shows the eigen-frequency of the first PMUT plate resonance according to the top electrode radius, which varies from 0 to 50 µm (100% surface metallization rate). The FDM predictions perfectly match the FEM simulations. The maximum relative error is 1.13% when the normalized electrode radius is close to 50%. When the electrode radius increases, a decrease in the resonance frequency is observed due to a mass effect; then, over a normalized electrode radius of 80%, the resonance frequency increases. This is ascribed to an increase in the global plate stiffness.

This first step clearly justifies the choice of modeling the silicon layer as an isotropic material and the numerical solving of the plate equation with discontinuity of the neutral line position.

### 2.3. PMUT/Fluid Coupling: Implementation of a Boundary Element Matrix

In this third subsection, the model is modified to introduce the coupling with a fluid, which is modeled through an external distributed pressure $q\left(r\right)=P_{0}(r)$. From plate Equation (11), one term must be added:(12)$$\left[K_{m}\right]\left[w\right]=\omega^{2}\left[M\right]\left[w\right]+\left[P_{0}\right],$$
where $\left[P_{0}\right]$ is the radiated pressure vector, which depends on the displacement vector $\left[w\right]$. The two vectors $\left[P_{0}\right]$ and $\left[w\right]$ are linked by the radiation boundary conditions at the PMUT/fluid interface. This relation can be expressed mathematically using a boundary matrix $\left[K_{fluid}\right]$:(13)$$\left[P_{0}\right]=\left[K_{fluid}\right]\left[w\right],$$
where in each term of the $\left[K_{fluid}\right]$ matrix, $K_{fluid\;i,j}$ corresponds to the mutual acoustic impedance between the *i*-indexed node and the *j*-indexed node. In other words, this term represents the force exerted by node *i* when it radiates with displacement $w_{i}$ onto node *j*. The radiating surface associated with node *i* is a ring centered at the $r_{i}$ position with Δ*r* width and the surface associated with node *j* is a ring centered at the $r_{j}$ with the same width. If the discretization pitch is thin enough, it is safe to assume that each ring vibrates like a piston. To compute the $\left[K_{fluid}\right]$ matrix terms, one has to use the Huygens–Rayleigh integral [63]:(14)$$\left[K_{fluid\;i,j}\right]=\int_{S_{j}}^{}dS_{j}\int_{S_{i}}^{}G\left(r_{i},r_{j}\right)dS_{i},$$
where $S_{i}$ and $S_{j}$ are the surfaces of the *i*-indexed ring and *j*-indexed ring respectively. $G\left(r_{i},r_{j}\right)$ is a Green’s function of the fluid for rigid baffle conditions:(15)$$G\left(r_{i},r_{j}\right)=\frac{ik_{0}\rho_{0}c_{0}}{2\pi }\frac{e^{-{ik}_{0}\left|r_{i}-r_{j}\right|}}{\left|r_{i}-r_{j}\right|},$$
where $k_{0}=\frac{\omega }{C_{0}}$ is the acoustic wavenumber in the fluid. $\rho_{0}$ and $c_{0}$ are acoustic properties (density and velocity, respectively) of the fluid. The numerical computation of the first integral (integration over the emitting ring of surface $S_{i}$) in Equation (14) can be extremely time-consuming because the convergence is typically very slow and strong numerical instabilities may be present. Seybert et al. [70] thus proposed splitting the integral into two parts to stabilize computations and decrease the convergence time. We have used this decomposition. The validation of the $\left[K_{fluid}\right]$ matrix terms’ computation was performed assuming that the PMUT vibrates as a piston, i.e., each term of the $\left[w\right]$ vector is equal to 1. The radiation impedance of the circular piston computed numerically with the FD model was compared with the analytical expressions [71]. No difference was observed.

### 2.4. Comparison with the Literature

For this second validation step, the analysis focuses on PMUT/fluid coupling, where the simulations obtained from the FD models are compared with those obtained with analytical models from the literature. The present goal is to discuss the limits of each approach and identify their respective advantages and disadvantages. Many analytical models have been published, but the two oldest [12,46] have been used as the basis for assumptions applied in the more recent works [33,59,61,72]. We have therefore chosen to focus our comparison on the model from K. Smyth et al. [46] and Perçin et al. [4,12].

Perçin [12] developed a lumped-element circuit to model PMUT/fluid coupling. From one side, the PMUT plate mechanical impedance is computed when the device is vibrating in air. A uniform external pressure source is applied over the whole plate, and the plate equation (Equation (4)) is solved to obtain the average plate velocity and thus the plate mechanical impedance. Analytical resolution of the plate equation based on modal decomposition of the solution is used. From another side, the plate radiation impedance of the PMUT is computed assuming that it vibrates like a piston with rigid baffle boundary conditions. The published radiation impedance expression of a piston in fluid [71] is directly used. K. Smyth et al. [46] developed the same approach to compute the plate mechanical impedance. The two models differ for the radiation impedance determination. K. Smyth et al. [46] took into account the nonuniform velocity distribution along the plate to compute the radiated pressure. The velocity distribution along the plate radius is the same as that obtained from the modal decomposition in air. This approximation is clearly better than considering that PMUT works like a piston, particularly at high frequencies where the velocity distribution impacts the PMUT directivity. Pala et al. [72] used the same method as Smyth except that a decomposition of quadratic-type functions was used instead of eigenfunctions of the plate equation.

To compare the three models, we used the device presented in Section 2.2 with a top electrode radius of 35 µm, so a metallization ratio radius of 70% was simulated. For the FD model, a uniform acoustic pressure of 1 Pa was applied (i.e., the $\left[P_{0}\right]$ vector in Equation (12)), and the spatial average plate velocity was computed from the computed displacement field vector [w]. For the analytical models, we implemented the analytical solutions provided by Smith [46] and Perçin [12] and computed the impedance terms. The average velocity is straightforward to determine from the lumped-element equivalent electrical circuit. Figure 5 shows the spatial average particle velocity of the PMUT according to the frequency for the three models in water, and the simulation of the device working for the FD model in air is added. The resonance frequency in air is close to 2.5 MHz, and it drops to a value close to 1 MHz in water. The three models yield very similar resonance frequency values in water, thereby validating the numerical implementation of the PMUT/fluid coupling in the FD model. However, the FD model differs from the two analytical models at higher frequencies. In air, at 9 MHz, a cutoff frequency appears that corresponds to the situation where the spatial average velocity of the plate is null. In water, with the FD model, we see that this cutoff frequency undergoes a mass-loading effect as it drops to 5 MHz, while the two analytical models provide 9 MHz, as in air (as if there was no interaction with a fluid medium). This cutoff frequency is a key characteristic of the PMUT cell because it also defines the high cutoff frequency of a PMUT array. This frequency remains the same whether the PMUT operates alone or as part of an array. This is clearly one of the limitations of the analytical models presented to date because the mechanical impedance of the PMUT plate is calculated when the device is vibrating in air. However, there are no simple analytical solutions to the plate equation when coupled to a fluid. It is therefore essential for the determination of equivalent mechanical and radiation impedances that the mechanical deformation of the plate used considers the coupling with water. This is the advantage (see Section 3.2) of having developed a numerical approach. However, it should be remembered from previous works, K. Smyth [46] and Pala [72], that when calculating radiation impedances, it is preferable to take plate deformation into account, particularly for the determination of the elementary directivity acoustic field but also the acoustic mutual impedances matrix between PMUTs. Though it is not further covered in this paper, additional investigations indicated that the type of pressure application and its distribution along the plate impact the cutoff frequency value as well. These additional investigations are ongoing and will be presented in future work.

## 3. Electroacoustic Modeling of a Single PMUT Cell Coupled with Fluid Medium

This third section completes the mechano-acoustic model of the PMUT cell in a fluid with piezoelectric coupling. The objective is to implement a lumped-element equivalent circuit of a PMUT cell from the set of numerically solved behavioral equations. The first subsection is devoted to the implementation of the piezoelectric coupling, and the second subsection is devoted to the equivalent electrical circuit determination. Note that in the first subsection, the impact of the axial displacement $u_{0}$ of the neutral line is discussed, and simulations are compared with the model developed by Sammoura et al. [35].

### 3.1. Implementation of the Piezoelectric Coupling

The piezoelectric coupling implementation needs to modify the equation of motion (Equations (4) and (12)) to introduce an additional source term. A second equation is needed, representing the balance of electrical charges produced by the PMUT, with electrostatic charges on one side and charges from piezoelectric coupling on the other.

For the equation of motion (4), the procedure has already been published [46], and the required source term is:(16)$$P_{p}\left(r\right)=\nabla^{2}M_{p}\left(r\right),$$
where the piezoelectric resultant moment $M_{p}\left(r\right)$ given by Smyth et al. [46] is:(17)$$M_{p}\left(r\right)=-\frac{E_{p}d_{31}V}{\left(1-\sigma_{p}\right)}\left(\frac{\left(h_{p}+h_{p-1}\right)}{2}-z_{n}\left(r\right)\right),$$
where $E_{p}$, $d_{31}$ and $\sigma_{p}$ are the Young’s modulus, the piezoelectric coefficient, and the Poisson’s ratio of the piezoelectric layer, respectively. $h_{p}$, as explained in the introduction of Section 2, refers to the upper *z*-coordinate (see Figure 1) of the piezoelectric layer and $h_{p-1}$ to its lower *z*-coordinate.

The same FD discretization scheme applied to the Laplacian operator (Equation (16)) enables the conversion of the piezoelectric source term $P_{p}\left(r\right)$ as an electrical-to-mechanical conversion matrix $\left[K_{em}\right]$ multiplied by the vector of the voltages $\left[V\right]$ associated with each discretization node. If the plate is fully metallized (100%), then all the nodes have the same voltage value, but when the plate is partially metallized (<100%), only nodes placed on the top electrode have the same voltage value. Outside the top electrode, the voltage node values are unknown while the electrical charge density is null. This represents a typical situation of mixed electrical boundary conditions. To simplify the computations next, we force the voltage outside the electrode to zero. Because the piezoelectric coupling of the plate is weak, the impact of such electrical boundary conditions on the electroacoustic response of the PMUT is low. In the case of a bulk piezoelectric plate, this assumption would be clearly not valid.

The equation of motion (12) becomes:(18)$$\left[K_{m}\right]\left[w\right]=\omega^{2}\left[M\right]\left[w\right]+\left[K_{em}\right]\left[V\right]+\left[P_{0}\right].$$

To establish the electrical charge balance, the expression for the electrical displacement $D_{z}$ in the piezoelectric layer is used [35]:(19)$$D_{z}\left(r\right)=d_{31}\frac{E_{p}}{\left(1-\sigma_{p}\right)}\left(S_{rr}+S_{\theta \theta }\right)+\varepsilon_{33}^{T}\left(1-k_{p}^{2}\right)E_{z},$$
where $\varepsilon_{33}^{T}$ is the permittivity at constant stress and $k_{p}^{2}$ is a coupling coefficient. Note that $k_{p}^{2}$ also includes the mechanical properties of the layer (Young’s modulus and the Poisson’s ratio):(20)$$k_{p}^{2}=\frac{2E_{p}d_{31}²}{\varepsilon_{33}^{T}\left(1-\sigma_{p}\right)}.$$

From the $D_{z}(r)$ expression, one can determine the surface electrical charge density expression [73]:(21)$$\sigma (r)=\int_{h_{p-1}}^{h_{p}}\frac{D_{z}}{{(h}_{p}-h_{p-1})}dz.$$

The plate is thin enough to consider $D_{z}(r)$ as constant along the *z*-axis, and the following relation for σ(r) is easily obtained from the Equations (19) and (21):(22)$$\sigma \left(r\right)=-\left(z-z_{n}\right)d_{31}\frac{E_{p}}{\left(1-\sigma_{p}\right)}\left(\frac{{𝜕}^{2}w}{𝜕r^{2}}+\frac{1}{r}\frac{𝜕w}{𝜕r}\right)\;+d_{31}\frac{E_{p}}{\left(1-\sigma_{p}\right)}\left(\frac{𝜕u_{0}}{𝜕r}+\frac{u_{0}}{r}\right)+\varepsilon_{33}^{T}\left(1-k_{p}^{2}\right)E_{z},$$
where $E_{z}$ is the electrical field along the *z*-axis. The electrical Equation (22) includes the axial displacement $u_{0}$ which is absent from some previous works [35,61,63] where it was neglected from the outset of the model design.

After applying the FD scheme, Equation (22) becomes:(23)$$\left[\sigma \right]=\left[K_{ee}\right]\left[V\right]+\left[K_{me}\right]\left[w\right]+\left[G_{me}\right]\left[u_{0}\right],$$
where $\left[K_{ee}\right]$ is the dielectric permittivity matrix which comes from the last term of (22), and $\left[K_{me}\right]$ is a mechanical-to-electrical conversion matrix coming from the first term of (22). The last matrix $\left[G_{me}\right]$ is also a mechanical-to-electrical conversion matrix that links the axial displacement vector $\left[u_{0}\right]$ with the electrical charge density. It is worth mentioning that $\left[K_{ee}\right]$ is a pure diagonal matrix because the electrical displacement $D\left(r\right)$ and the electrical field $E\left(r\right)$ are assumed to be oriented along the *z*-axis only.

To compute $\left[\sigma \right]$, $\left[u_{0}\right]$ must be determined first by using Equation (5) and Equation (A17) given in the Appendix B Section. Equation (A17) gives the equation of motion associated with the resultant stresses (5) with the piezoelectric coupling.

Equation (A18) is easy to solve numerically for a given excitation input voltage vector $\left[V\right]$. To avoid burdening the body of this article, all the equations for determining $\left[u_{0}\right]$ are reported in Appendix B.

The electrical current I flowing through the PMUT is obtained after numerical integration of the charge density:(24)$$I=2\pi \int_{0}^{r_{e}}\sigma \left(r\right)r\;dr.$$

Having determined the relation between the current and the voltage, we can deduce the electrical admittance, the electrical impedance, and the parallel capacitance.

Before presenting simulation results, and discussing the impact of $u_{0}$, we can compare the two equations we have to solve (10) (equation of motion) and (22) (electrical equation) with the ones presented by Sammoura et al. [35]. Since Sammoura considered the influence of axial displacement on the transverse displacement, $\left[w\right]$, can be neglected and the two equations of motion (10) are the same. On the other hand, in the electrical equation, Sammoura introduces a second neutral line *z_e_*, so-called the electrical neutral line, to compute the resultant piezoelectric force. Analogy with the Equation (22) given above suggests that this electrical neutral line may be interpreted as the impact of axial displacement $u_{0}$ on the electrical charge balance.

We implemented Sammoura’s model and compared it with the FD model considering $u_{0}$ and without considering $u_{0}$. Electrical impedance was simulated for the device presented in Section 2.4 (with a radius of the top electrode of 35 µm, equivalent to 70% of the PMUT surface). Figure 6 shows the real part of the electrical impedance for water loading conditions with $\rho_{0}$ = 1000 kg/m^3^ and $c_{0}$ = 1500 m/s and the equivalent parallel capacitance obtained from the admittance. The three models are very similar with an error less than a few percent. The low-frequency capacitance is 0.662 pF for the FD model with $u_{0}$, 0.664 pF for the FD model without $u_{0}$, and 0.684 pF for Sammoura’s model, corresponding to a 3% relative error. Note that if the piezoelectric layer was PZT-based (with a stronger piezoelectric coupling coefficient), the difference between the FD model and Sammoura’s model should be slightly larger but not significant. Finally, as a partial conclusion, the comparison of simulations with and without $u_{0}$ clearly shows that the impact of axial displacement on the electrical charge density is very weak and can be reasonably neglected for the rest of the paper. The simulation with PZT-based PMUT leads to the same conclusion. This hypothesis is a common practice in the prior reports, based on many authors having adopted it. However, they never clearly concluded whether the role of $u_{0}$ was important for the simulation of the PMUT electroacoustic response. We consider that the current scope of the literature is ambiguous around which assumptions should or should not be considered to construct an analytical model of PMUT.

### 3.2. Equivalent Lumped-Element Model Implementation

From the set of Equations (18) and (23), assuming $u_{0}$ to be negligible, the electroacoustic response of a PMUT can be totally modeled in the transmit and receive modes. However, to compute the response of an array of PMUTs, the equivalent lumped-element electrical circuit of a single PMUT must be determined first to reduce the number of degrees of freedom in the simulation. All analytical models previously developed lead to the construction of this equivalent circuit. The lumped-element equivalent electrical circuit scheme that we propose is presented in Figure 7. This is the same as that presented in previous works, whether for PMUT [46,59,60,61] or CMUT [42,74].

When working as an emitter, the input excitation sources are the electrical voltage *V* and the electrical current *I*, and the output acoustical data are the spatial average plate velocity *<*$\dot{w}$*>* and the radiated acoustic force $F_{r}$. The equivalent circuit comprises five parameters:parallel electrostatic plate capacitance $C_{0}$,equivalent mechanical impedance of the plate $Z_{ms}$,radiation impedance $Z_{r}$,electrical-to-mechanical transformation factor $\Phi_{em}$,mechanical-to-electrical transformation factor $\Phi_{me}$.

We further mention that two transformation factors are voluntarily introduced because, looking at the terms of the $\left[K_{em}\right]$ and $\left[K_{me}\right]$ matrices, it is not immediately clear that the two transformation factors are identical. This is not the case for CMUT, where both matrices are made with the same terms, provided that the electrostatic edge effects are neglected [41].

To determine the five parameters listed above, we define which quantities are maintained from the distributed FD model to the lumped-element model. It is straightforward to assume that *I* and *V* will be the same. On the acoustic port side, the spatial average velocity *<*$\dot{w}$*>* is also maintained. For the second acoustic quantity, there is a debate; some authors prefer keeping the radiation force [12,46,72] (so known as Mason’s model) whereas others prefer keeping the acoustic energy [42,74] as defined by Foldy [75,76]. We made this choice because Foldy’s definition enables us to directly obtain the time average radiated power by the transducer, as explained by Sherman’s book [71]. The time average radiated power is an essential feature of ultrasonic transducers, both for imaging and high-intensity applications. To compute the radiated pressure, the average velocity is taken as a radiating source condition in the Rayleigh integral.

The numerical determination of these quantities requires two steps: first, to run the distributed FD model for a given input source voltage, typically 1 Volt, and then, to compute the acoustic power stored in the plate (the mechanical power $W_{m}$) and the radiated acoustic power ($W_{r}$). As explained in [74], the two impedance terms and the transformation factors are written as:(25)$$Z_{ms}=\frac{2W_{m}}{\left|{\left\langle \dot{w}\right\rangle }^{2}\right|},$$
(26)$$Z_{r}=\frac{2W_{r}}{\left|{\left\langle \dot{w}\right\rangle }^{2}\right|}.$$
(27)$$\emptyset_{em}=\frac{2\left(W_{m}+W_{r}\right)}{V{\left\langle \dot{w}\right\rangle }^{*}},$$
(28)$$\emptyset_{me}=\frac{2{We}^{*}}{\left\langle \dot{w}\right\rangle V^{*}}.$$

$C_{0}$ is deduced from the integral of the $\left[K_{ee}\right]$ matrix diagonal terms.

The electrical admittance of the PMUT is obtained as:(29)$$Y_{elec}=j\omega C_{0}+\frac{\emptyset_{em}\emptyset_{me}}{Z_{ms}+Z_{r}}.$$

To conclude this section, we note that our approach is very similar to previously published works. However, there is an essential difference: the calculations of the mechanical and acoustic impedances are carried out jointly and come from a vibratory solution where the PMUT vibrates in water. This enables the correct prediction of the resonance center frequency of the PMUT, the high cutoff frequency, and all the other axisymmetric modes.

## 4. Experimental Validation

In this last section, validation of the model is completed by comparing simulations with experimental results obtained from an AlN-based PMUT. After a brief description of the tested device, electrical impedance and displacement measurements are performed in air and in a vegetable oil. The corresponding results are presented and discussed in view of the simulation results.

### 4.1. Device Description

The tested PMUT was fabricated by MEMSCAP Inc (Grenoble, France). The fabrication process flow (Figure 8) follows the steps of the PiezoMUMPS process [77]. The process starts (Figure 8a) with a Silicon On Insulator (SOI) wafer made with a 1 µm oxide layer and a doped silicon layer of 10 µm. The silicon layer is the structural layer of the PMUT plate, and the thickness can be tuned according to the targeted working frequencies. The doped silicon layer also plays the role of the bottom electrode. (Figure 8b) An insulating layer based on thermal oxide (200 nm thick) is then created and patterned by wet etching. The oxide layer is required to insulate the AlN layer from the bottom electrode everywhere except at the cavity location and the pad of the bottom electrode. Then, (Figure 8c) 500 nm of piezoelectric aluminum nitride is deposited by reactive ion sputtering on the whole wafer surface. Vias are etched to reveal the bottom electrode. The last layer deposited is the metallization layer (Figure 8d), which is composed of 20 nm of chrome and 1 µm of aluminum and is patterned through a liftoff process. The metallization layer is used for the top electrode and for the contact pads of the electrical ground. After protecting the front side with a polyimide coating (Figure 8e), the bottom side is etched using Reactive Ion Etching (RIE) to pattern the cavities of the cells (Figure 8f). The last step (Figure 8g) is the protection layer removal with a dry etching process.

For model validation, two PMUT cell sizes were investigated: one with a diameter of 340 µm and the second with a diameter of 275 µm. These PMUT sizes may seem large regarding the final applications targeted (typically medical imaging). However, constraints from the process flow did not enable us to fabricate PMUT with diameter smaller than 200 µm. To counterbalance these PMUT cell sizes, the structural layers of the plate were designed to reach a working frequency in the MHz range. Both cells are presented in Figure 9, where each top electrode is linked to the metal pad (250 × 100 µm^2^) by a track of 50 µm width. Two metal pads are connected to the bottom electrode spaced on other sides of the top metal pad at 100 µm. The cavities have been represented by adding a circle in dashes.

### 4.2. Experimental Results and Discussion

Electrical impedance measurements with a 4294A impedance analyzer (Agilent Technologies Inc., Santa Clara, CA, USA) were performed in air first. The input data used for AlN, Si, and Al to simulate the devices are recalled in Table 2. Chrome parameters are added in Table 2. A mechanical loss factor was also added in the stiffness matrix $\left[K_{m}\right]$ to fit the experimental quality factor. Figure 10 shows the real part of impedance (Figure 9a) and the equivalent parallel plate capacitance (Figure 9b) measured for the second device (275 µm) and the simulated device. Theoretical results are in good agreement with experiments; the same approach was carried out with the first device 1 (340 µm) by changing the dimensions, and the conclusions are identical with a very good agreement between theory and experiment.

Electrical impedance measurements were recorded when the devices vibrate in a fluid, thus completing the experiment and enabling the validation. Oil was chosen rather than deionized water to minimize capacitive coupling between the bottom and the top electrodes at the PMUT surface. All the device parameters used for air were kept for this second set of simulations in vegetable oil (Table 2). The acoustic oil properties used were an ultrasonic velocity of 1450 m/s and a density of 920 kg/m^3^. Figure 11a,b show the real part of the impedance and the parallel plate capacitance for the first PMUT (diameter 340 µm), respectively. Figure 11c,d show the real part of the impedance and the parallel plate capacitance for device 2 (diameter 275 µm), respectively. For device 1 and device 2, the resonance frequencies decreased from 1.5 MHz to 0.69 MHz and from 2.3 MHz to 1.15 MHz, respectively.

The FD model perfectly matches the experimental results. In particular, the resonance frequencies are the same. We mention that the quality factor is also accurately predicted. This is a key validation because the PMUT damping is mainly due to ultrasonic wave radiation and not due to mechanical losses. This means that the PMUT/fluid coupling is well accounted for by the FD model. It is worth noting that the asymmetrical vibration condition matches the experimental results even if the PMUT topology shows a slight asymmetry due to the metallization strip required to have the top electrode electrical contact. Finally, the mechanical displacement of the two PMUT cells was measured with a laser vibrometer (UHF-120, Polytec GmbH, Waldbronn, Germany). The two devices were placed in vegetable oil, and the displacement was scanned to compute the experimental spatial average displacement. A 2D cartesian sweeping with a pitch of 10 µm was realized. The sweeping zone dimensions were chosen to cover all the electrode surfaces. To compute the spatial averaging, only points that belong to the circular surface of the top electrode were kept. The excitation signal used was a monopolar sinus-shaped pulse of 40 V amplitude and 5 MHz center frequency. Such an excitation signal enables us to cover all frequencies up to 10 MHz and thus potentially to observe the resonance frequency of the PMUT cell as well as its high cutoff frequency. The results are presented in Figure 12. In the time domain, the displacement response at the plate center is presented for the two configurations. The simulation results perfectly match the experiments, and even the signal damping seems good, which agrees with the conclusion obtained from the impedance measurements. For the displacement amplitude, the difference is ascribed to the variation in the optical index between oil (1.47) and air (1) and acousto-optic interactions [78]. The measured displacement amplitude should be divided by 1.47 if acousto-optic interactions are weak. If this factor is applied to the experimental displacement data, it is clear that the difference between the theory and experiment is minimal.

From the experimental displacement spectra, the frequency response of the spatial average PMUT velocity was numerically computed. The results are presented in Figure 13a,b. The maximum value for the 340 µm cell is at 0.7 MHz, and that for the 275 µm cell is close to 1.2 MHz. These values are consistent with the resonance peak observed for the experimental electrical impedance responses. The theoretical spatial average velocity is added to the experimental curves. The theoretical resonance frequencies and damping factors agree very well with the experimental values. As predicted by the model, the cutoff frequency is strongly impacted by the fluid-loading effect as well. In air, for the 340 µm diameter PMUT, the value was 5.75 MHz and this shifts to 4 MHz in a fluid. For the 275 µm PMUT diameter, the cutoff frequency value shifted from 8.8 MHz to 6 MHz. FD model cutoff frequency values perfectly match with experimental ones. As explained previously, this specific frequency defines the upper frequency above which the whole PMUT plate does not vibrate in phase. The shape of the PMUT deformation for the resonance and the cutoff frequencies were investigated and compared with theory. For the resonance frequency, all of the surface of the PMUT plate vibrates in phase. However, as expected, for the cutoff frequency, the outer part of the plate is out-of-phase with the inner part, which leads to a null spatial mean velocity. If the PMUT was surrounded by an array of identical PMUT, the cutoff frequency would be the same. This last validation point clearly justifies our choice of using a numerical model to compute the lumped-element equivalent circuit of a single PMUT cell. This result has never been noted to date.

## 5. Conclusions

Here, we develop an analytical model of a single PMUT cell vibrating in a fluid and construct an equivalent lumped-element electrical circuit. These steps are necessary to simulate entire PMUT arrays comprising dozens or even hundreds of simultaneously vibrating PMUTs. Of course, this is not a new challenge, because many authors have already proposed relevant analytical approaches capable of predicting, with reasonable accuracy, the electroacoustic response of a PMUT cell radiating in water. In such works, developing the analytical model requires simplifying the behavioral equations and making several assumptions, due to which analytical solutions for PMUT systems can be established. However, many assumptions that have been made toward developing analytical solutions are sometimes faced with ambiguous validity. Here, we attempted to test these assumptions in parallel with the stepwise development of our model.

The outcome shows that silicon material can be modeled as an isotropic material and that a PMUT with a circular topology can be described under axisymmetric conditions. We have demonstrated an expression for the position of the neutral line and shown how to simplify, without any assumptions, the behavioral equations of a plate to yield one equation of motion that is associated with transverse displacement and a second equation that is associated with axial displacement. This demonstration is only applicable if the materials are isotropic or transversely isotropic, which is often the case for MEMSs. We have also confirmed a result that is widely accepted in the literature: axial displacement can be neglected and has no significant impact on the electrical behavior of the PMUT.

To solve the system of equations, we do not develop analytical solutions for two reasons. The first is attributed to the inhomogeneous nature of the plate, whose thickness varies, and the second is attributed to solving the fluid/structure coupling equations. Thickness variation results in a neutral line discontinuity that is difficult to introduce in a set of analytical solutions. Regarding the second reason to avoid analytical solutions, prior work has shown that fluid/structure coupling can only be correctly simulated using an eigenmode solution basis associated with the governing equation of a fluid-loaded plate. However, many current analytical models are based on the eigenmodes of the plate vibrating in air. We thus chose to use a finite difference discretization scheme. This type of discretization is easy to implement and not very demanding in terms of computational time and capacity. The accuracy of the model has been validated in both air and fluid. To integrate fluid/structure coupling, a boundary element matrix for an axisymmetric radiation source has been developed, implemented, and validated.

For the implementation of the equivalent lumped-element electrical circuit, the solutions take fluid/structure coupling into account from the start, enabling the correct prediction of the PMUT’s center frequency and high cutoff frequency. Moreover, Foldy’s definition was used, based on power conservation, whereas Mason’s model (based on force conservation) is the most common for PMUT. It seems that both approaches are fair and that the choice is open.

Finally, the model developed was experimentally validated using AlN-based PMUTs. Electrical and mechanical measurements were carried out in a fluid medium. The theory–experiment comparison validates the model and confirms all the assumptions we have made regarding the analytical models published to date.

This work will later be extended to the modeling of PMUT arrays to be able to simulate all the basic characteristics of an ultrasonic transducer, electroacoustic response, center frequency, bandwidth, etc. The choice of source conditions to be used to determine the matrix of mutual impedances between PMUTs will be an important point to address in view of the work presented by K. Smyth et al. [46].

## Figures and Tables

**Figure 1 micromachines-14-02089-f001:**
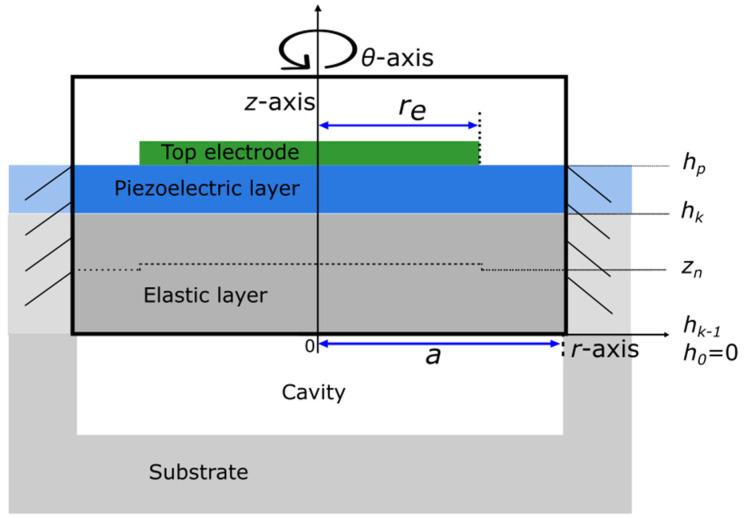
Cross-sectional view of axisymmetric circular PMUT with top partial electrode coverage. The stack is made up, from bottom to top, of a structural layer, a bottom electrode, a piezoelectric layer, and a partially covering top electrode. The terms a and $r_{e}$ are the plate radius and the top electrode radius, respectively.

**Figure 2 micromachines-14-02089-f002:**
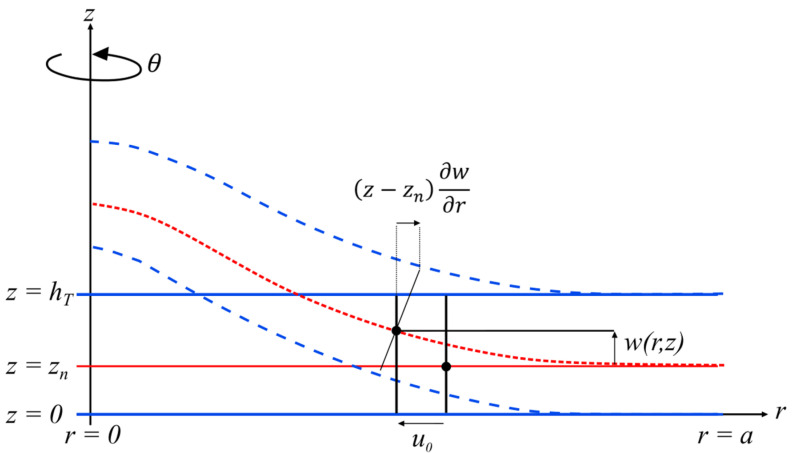
Total displacements in an axisymmetric plate with 100% of metallization (from M. Geradin, Chapter 4 [56]).

**Figure 3 micromachines-14-02089-f003:**
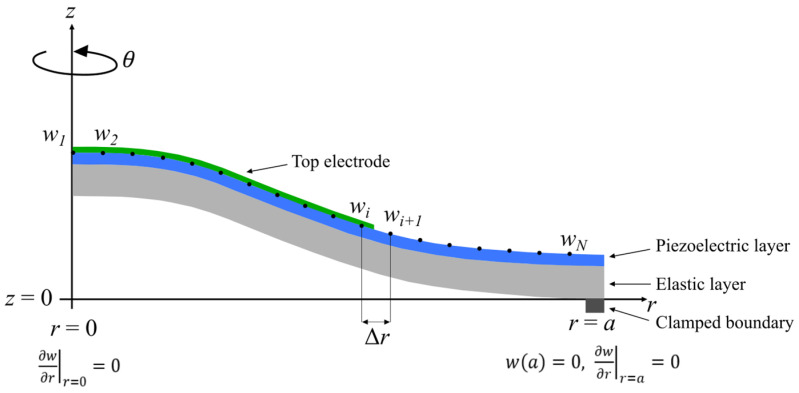
Discretization nodes along the radius of the plate with a partially recovering top electrode.

**Figure 4 micromachines-14-02089-f004:**
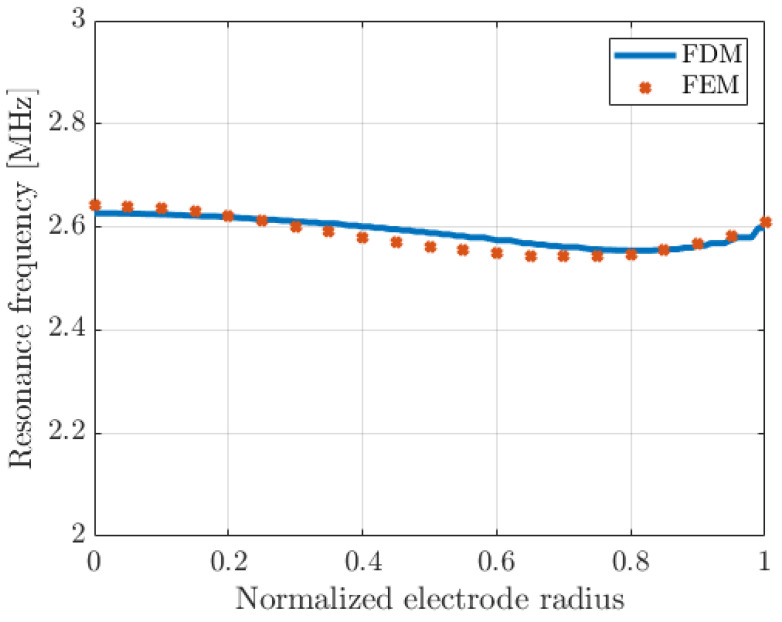
Variations in the resonance frequency of the first plate mode according to the top electrode radius: comparison between FDM and FEM. The top electrode radius is normalized by the PMUT radius, i.e., 50 µm.

**Figure 5 micromachines-14-02089-f005:**
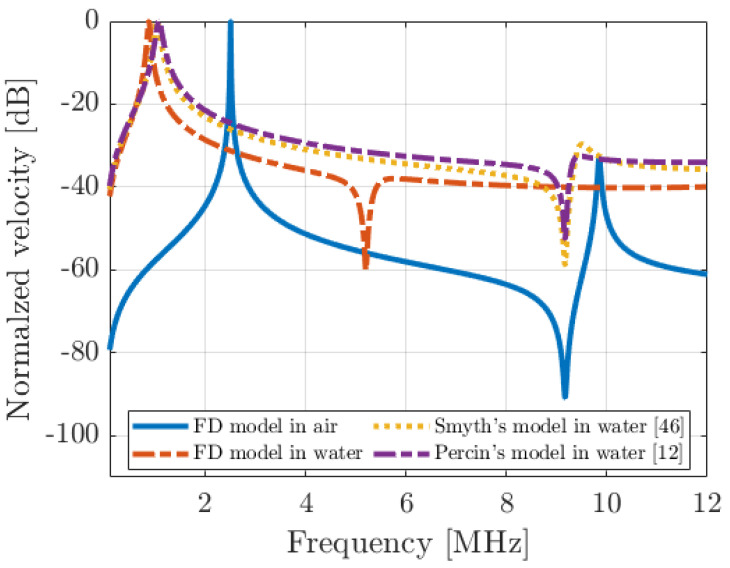
Spatial average particular velocity of the PMUT described in Table 1 (radius top electrode of 35 µm—70%) for 1 Pa of pressure applied, and comparison between the FD model in air and in water with the model from Smyth, 2015 and the model from Perçin, 2002.

**Figure 6 micromachines-14-02089-f006:**
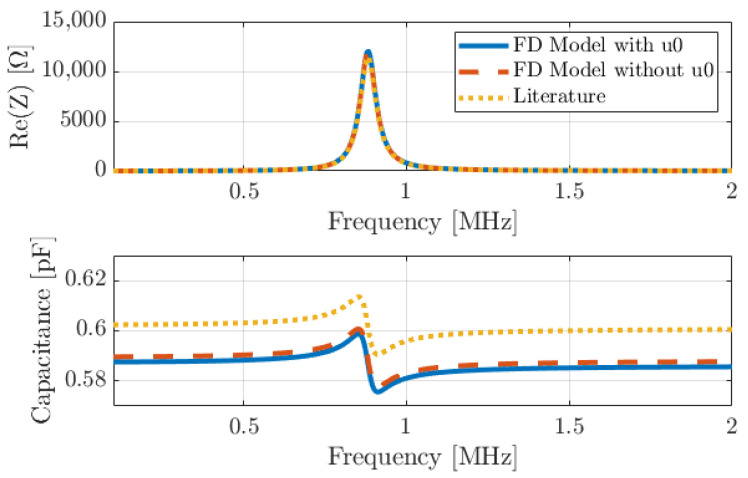
Theoretical real part of electrical impedance and capacitance for the plate described in Table 1 (0.7), with comparison of our model with and without considering the axial displacement ($u_{0})$ and Sammoura’s model (Literature) for AlN as a piezoelectric material in water.

**Figure 7 micromachines-14-02089-f007:**
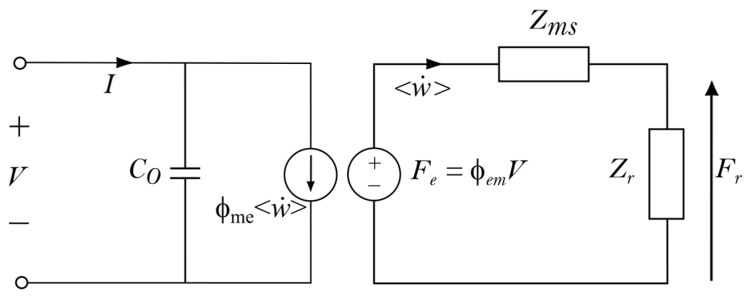
Lumped-element electrical circuit of a single PMUT vibrating in fluid.

**Figure 8 micromachines-14-02089-f008:**
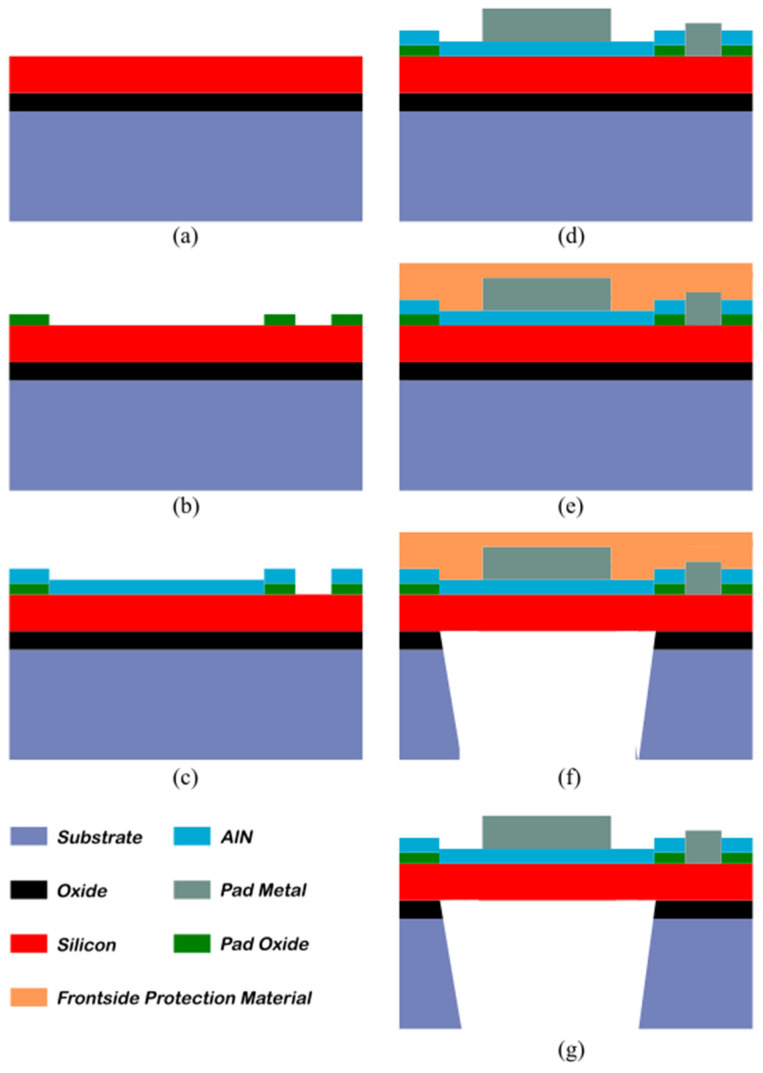
The fabrication process flow of the PMUT [65]. (**a**) SOI wafer made with the oxide layer and doped silicon layer. (**b**) Deposition and patterning of the thermal oxide layer used to insulate the AlN layer from the bottom electrode. (**c**) AlN layer deposition and vias etching to access the bottom electrode. (**d**) Deposition and patterning of the metallization layer, used for the bottom electrode and the ground pads. (**e**) Polyimide protection layer before RIE etching of the cavities. (**f**) Cavity etching. (**g**) Removing the protection layer.

**Figure 9 micromachines-14-02089-f009:**
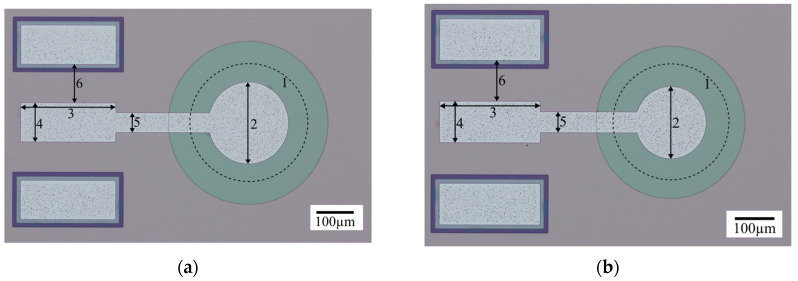
Optical microscope photographs of the two devices used for the comparison between the experimental and theoretical results. (**a**) Cell of 340 µm diameter (cavity in dash 1) with a top electrode of 210 µm diameter (2). (**b**) Cell of 275 µm diameter (cavity in dash 1) with an electrode of 173 µm diameter (2). The dimensions of (3), (4), (5), and (6) are 250 µm, 100 µm, 50 µm, and 100 µm, respectively.

**Figure 10 micromachines-14-02089-f010:**
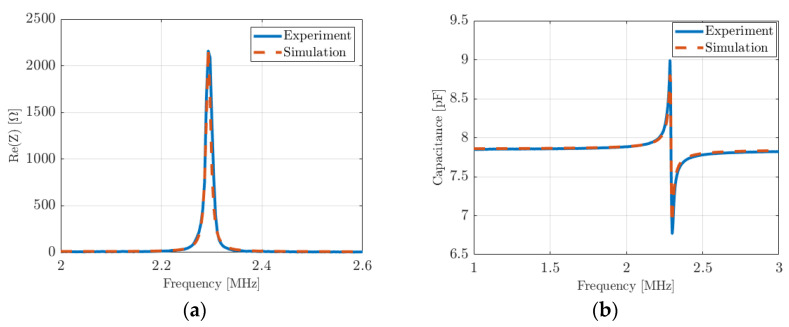
Electrical impedance of a PMUT cell vibrating in air (275 µm diameter): (**a**) real part and (**b**) equivalent parallel plate capacitance. Red-dashed curves are simulations and blue curves are measurements.

**Figure 11 micromachines-14-02089-f011:**
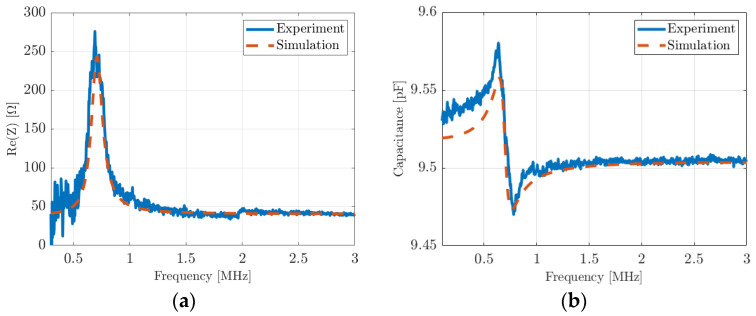
Electrical impedance of a PMUT cell vibrating in oil: (**a**) the real part of the first device (340 µm diameter) and (**c**) the second device (275 µm diameter); (**b**) the equivalent parallel plate capacitance for the first device (340 µm diameter) and (**d**) the second device (275 µm diameter). Red-dashed curves are simulations, and blue curves are measurements.

**Figure 12 micromachines-14-02089-f012:**
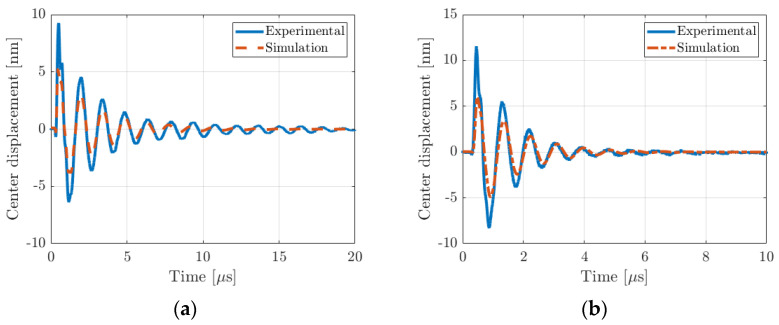
Time response of the center displacement for the two devices: (**a**) 340 µm diameter and (**b**) 275 µm diameter. Red-dashed curves are simulations, and blue curves are measurements. Electrical excitation was monopolar sinus-shaped (40 V amplitude and 5 MHz center frequency).

**Figure 13 micromachines-14-02089-f013:**
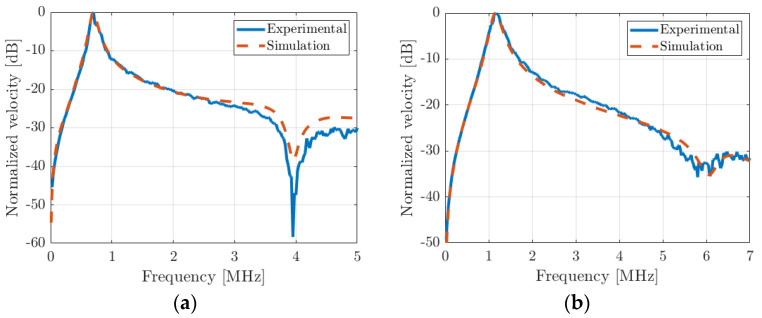
Frequency response of the spatial average particular velocity for the two devices: (**a**) 340 µm diameter and (**b**) 275 µm diameter. Red-dashed curves are simulations, and blue curves are measurements. Excitation was monopolar sinus-shaped (40 V amplitude and 5 MHz center frequency).

**Table 1 micromachines-14-02089-t001:** The material dimensions and material properties used for simulations of the Al top electrode radius can vary between 0 and 50 µm, corresponding to 0 to 100% of the cavity radius.

Layer	Si [68]	AlN [69]	Al [69]
Thickness [µm]	1	0.5	0.2
Radius [µm]	50	50	0–50
Young’s Modulus [GPa]	143	348	69
Poisson’s ratio	0.28	0.24	0.35
Density [kg/m^3^]	2330	3600	2700
d_31_ [pC/N]	-	−2.05	-
$\varepsilon_{33}^{T}/\varepsilon_{0}$	-	10	-

**Table 2 micromachines-14-02089-t002:** Material dimensions and properties used for simulations.

Layer	Si [68]		AlN [69]		Cr		Al [69]	
Device	1	2	1	2	1	2	1	2
Diameter [µm]	340	275	340	275	211	173	211	173
Thickness [µm]	10		0.5		0.2		1	
Young’s modulus [GPa]	143		348		279		69	
Poisson’s ratio	0.28		0.24		0.21		0.35	
Density [kg/m^3^]	2330		3600		7190		2700	
d_31_ [pC/N]	-		−2.05		-		-	
$\varepsilon_{33}^{T}/\varepsilon_{0}$	-		10		-		-	

## Data Availability

Data are contained within the article.

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
