# Peer review of "Modeling a Fluid-Coupled Single Piezoelectric Micromachined Ultrasonic Transducer Using the Finite Difference Method"

_micromachines, 2023, doi:10.3390/mi14112089_

Round 1

Reviewer 1 Report

Comments and Suggestions for Authors

This paper presents a computational method for the analysis of Piezoelectric Micromachined Ultrasonic Transducers, which are modeled as circular plates. The equations of motion are derived based on classical plate theory and solved utilizing a finite difference approach. The outcomes of this work are compared to results from COMSOL, those from the literature, and experimental outcomes, showing good agreement between the results. 

In summary:

- The title of the paper is of interest.

- The introduction describes the background of the work well.

- The formulation is clear, and the outcomes are compared step by step to FEM outputs, results from the literature, and experimental observations.

In my opinion, the paper is well-qualified for publication. However, the authors may consider the following suggestions: 

1- Section 2, equations 1-19: The procedure for deriving equation 17 can be found in several books on the mechanics of plates, including those referred to by the authors. This part can be shortened, or some equations can be included in the appendix to make the paper more concise.

2- On page 8, in the third paragraph, the finite difference (FD) approach could be discussed in more detail since it is a notable part of the proposed solution. For example, the number of nodes considered on the plate or any convergence considerations for the FD method could be discussed.

Author Response

Please see the attachment - the revised document is color marked to highlight the corrections.

Reviewer 2 Report

Comments and Suggestions for Authors

Accurate calculation of coupling between transducer and surrounding fluid can help researchers and engineers better understand the vibration mechanism of transducers in actual working conditions.

The most interesting findings the authors have found in the paper are the differences of vibration properties of transducers in the fluid and in the air, such as the decrease of higher-mode cutoff frequencies. The phenomenon is worth discussing.

(1) As illustrated in Fig 5, the resonance frequency of the fundamental mode of transducer reduces from 2.5MHz to 1.0MHz when merged in water than in the air, as has been pointed out by authors and previous studies. Wheareas for higher vibration modes, the more obvious resonance frequency reduction (from 9.0MHz to 5.0MHz) is suggested by current paper but not mentioned by previous models. However, the authors have not elaborated on this phenomenon. Can this frequency be validated by FEM simulation or experimental procedures? Also, can the vibration displacement shapes be plotted?

In later sections, the experiments seem to be focused on other modes. The reviewer suggest the authors to explain about the 5.0MHz vibration mode since it is more important in application.

(2) In calculation of transducer-fluid interaction, the authors mentioned usage of Green’s function. The explanation in Section 2.3 could be more detailed. Are the points ri and rj points within the fluid media or points on the transducer surface? What is the source term in Eqn. (23)? Is the source term body force or displacement or other quantities? (It seems to be a uniform 1 across the plate.) If ri and rj points are on the transducer surface, can Eqn. (23) and (24) directly used, since now the points become boundary points from viewpoint of surrounding fluid media?

(3) The word “increase” should be “decrease” in line 377. The vector [q] should better replaced by [p0] in line 410 for consistency.

Author Response

(The authors gave the same response as above.)

Reviewer 3 Report

Comments and Suggestions for Authors

I have reviewed the manuscript carefully entitle “Modeling a fluid coupled single PMUT using the finite difference method”. The topic is useful and interesting. My concerns are given below.

·       Abstract is too general and needs revision with pacific more numeric information.

·       What about the fluid structural interaction for electrostatic/fluid.

·       Model is presented for Piezoelectric Micromachined Ultrasonic Transducer (PMUT), but in manuscript there is no correlation with it.

·       On what ground the material dimensions and material properties used in table was taken? Same is the case for table 2.

·       As discuss the mechanical behavior? What about stress, stiffness matrix?

·       Authors need to explain compression and expansion during piezoelectric coupling,

·       Authors need to be clear  Piezoelectric transducer and electrostatic equivalent model, need to discuss correlation.

·       Manuscript is lack of focus discussion and authors presented results but focused discussion is missing.

·       Bench mark table should be added by giving the compression of current study with literature by giving its pros and cons with previous studies.

Comments on the Quality of English Language

looks ok

Author Response

(The authors gave the same response as above.)

Reviewer 4 Report

Comments and Suggestions for Authors

A good work! Scientifically interesting, well organized and clearly presented. My opinion is that it can be published.

Author Response

(The authors gave the same response as above.)

Round 2

Reviewer 3 Report

Comments and Suggestions for Authors

Revisions are satisfactory therefore recommend it for publication